# SERS Investigation on Oligopeptides Used as Biomimetic Coatings for Medical Devices

**DOI:** 10.3390/biom11070959

**Published:** 2021-06-29

**Authors:** Michele Di Foggia, Vitaliano Tugnoli, Stefano Ottani, Monica Dettin, Annj Zamuner, Santiago Sanchez-Cortes, Daniele Cesini, Armida Torreggiani

**Affiliations:** 1Dipartimento di Scienze Biomediche e Neuromotorie, Università di Bologna, 40126 Bologna, Italy; vitaliano.tugnoli@unibo.it; 2Istituto per la Sintesi Organica e la Fotoreattività, Consiglio Nazionale delle Ricerche (ISOF-CNR), 40129 Bologna, Italy; stefano.ottani@isof.cnr.it (S.O.); armida.torreggiani@isof.cnr.it (A.T.); 3Dipartimento di Ingegneria Industriale, Università di Padova, 35131 Padova, Italy; monica.dettin@unipd.it (M.D.); annj.zamuner@unipd.it (A.Z.); 4Instituto de Estructura de la Materia, Consejo Superior de Investigaciones Cientificas (CSIC), 28006 Madrid, Spain; s.sanchez.cortes@csic.es; 5Dipartimento CNAF, Istituto Nazionale di Fisica Nucleare (INFN-CNAF), 40127 Bologna, Italy; daniele.cesini@cnaf.infn.it

**Keywords:** amphiphilic oligopeptides, SERS, biomimetic coating, DFT, oligopeptide–surface interaction, oxidative stress

## Abstract

The surface-enhanced Raman scattering (SERS) spectra of three amphiphilic oligopeptides derived from EAK16 (AEAEAKAK)_2_ were examined to study systematic amino acid substitution effects on the corresponding interaction with Ag colloidal nanoparticles. Such self-assembling molecular systems, known as “molecular Lego”, are of particular interest for their uses in tissue engineering and as biomimetic coatings for medical devices because they can form insoluble macroscopic membranes under physiological conditions. Spectra were collected for both native and gamma-irradiated samples. Quantum mechanical data on two of the examined oligopeptides were also obtained to clarify the assignment of the prominent significative bands observed in the spectra. In general, the peptide–nanoparticles interaction occurs through the COO^−^ groups, with the amide bond and the aliphatic chain close to the colloid surface. After gamma irradiation, mimicking a free oxidative radical attack, the SERS spectra of the biomaterials show that COO^−^ groups still provide the main peptide–nanoparticle interactions. However, the spatial arrangement of the peptides is different, exhibiting a systematic decrease in the distance between aliphatic chains and colloid nanoparticles.

## 1. Introduction

In the field of functional biomaterials, peptides and oligopeptides can provide several advantages at the nanoscale, mainly related to their high biocompatibility, cell permeability, and low immunogenicity [1,2,3]. The 20 natural L-amino acids can be assembled in vast numbers of combinations to encompass a massive range of properties, making them suitable for applications in entirely different fields, such as hydrogels for extracellular matrix and hybrid materials for biosensing.

The study of the sequences of yeasts’ proteins led to the development of synthetic materials promoting cell growth, composed of regularly alternating polar/nonpolar amphiphilic oligopeptides, whose progenitor was EAK16 (AEAEAKAK)_2_, first synthesized by Zhang and co-workers [4,5,6]. These molecular systems display complementary polar surfaces, viz. two hydrophilic surfaces can interact by positively and negatively charged amino acid residues at physiological pH, which complement each other, favoring the establishment of hydrophobic interactions, dipole electrostatic forces, π-π stacking, and hydrogen bonding. As a result, these compounds are observed to self-assemble into unusually stable β-sheet structures [4,5,7], giving rise to insoluble macroscopic membranes under physiological conditions, typically favored by monovalent cations [4]. Since their discovery, these systems have also been known as “molecular Lego”. In fact, Lego bricks can be assembled only by matching specific sides, a hole-side with a peg-side, similarly to the behavior of these peptide systems where interactions between complementary polar surfaces give rise to remarkably stable secondary structures.

Such self-assembling oligopeptides have shown to possess chemical and physical stability, as confirmed by their resistance to heat and denaturation by several chemical agents and enzymes [4]. They can be easily fabricated in different geometrical shapes [5,7], including vesicles, spherical or elongated micelles, and nanotubes. Moreover, several studies show their ability to provide stable attachments with mammalian cells, supporting cell proliferation and differentiation [8,9,10,11]. This set of properties is especially relevant in advanced biotechnological applications, particularly in the nanofabrication of biomedical devices for bone tissue engineering, constituted by TiO_2_ surfaces supporting self-assembled peptide layers.

Two main points are essential for the materials’ long-term applications: the possible chemical modifications of the peptides and the stability of the corresponding self-assemblies under different conditions [12]. Since the inflammatory processes play a crucial role in the early stages of implanting a biomedical device into the body, the reactions at the interface between the biomaterial and the surrounding tissues can strongly affect the success of an implant [13,14]. Thus, it is essential to evaluate the structural changes induced in these oligopeptides by the sequence modifications and the interactions with the biological environment containing metal nanoparticles (NPs) [15,16] or •OH radicals. Free radicals are constantly formed in the human body during cell growth and in chronic inflammation [13,14], and, even if the EAK peptides reduce this inflammatory reaction [7], it cannot be excluded that •OH radicals at high concentrations alter the biomaterials.

Raman spectroscopy has proven to be an advantageous technique to investigate oligopeptides’ structure [17,18,19,20], but two limitations have to be considered. Firstly, most biomolecules display intrinsic fluorescence, which in some cases overwhelms the intensity of the Raman signal; secondly, this technique has a relatively low sensitivity in aqueous solutions, such as the physiological environment. Both drawbacks can be overcome using the SERS technique (Surface Enhanced Raman Scattering), based on the studied molecules’ interactions with NPs. In SERS experiments, fluorescence is usually quenched, and very low detection limits in a solution can be achieved (up to 10^−15^ M in selected cases, such as Rhodamine 6G and 10^−8^ M for peptides) [21,22]. Such an enhancement may allow the study of peptides structure at the interface between the biomaterial and body fluids. It might also provide a test for the presence of oligopeptides in the aqueous environment surrounding a metal implant. Besides, information on the packing and orientation (adsorbed structure) of molecules on the metal surface can be obtained by SERS spectra [23] because specific selection rules take place on the surface vibrations (i.e., more significant tensor components oriented along the vertical axis to the metal surface will undergo a higher enhancement due to the larger field in this direction) [24].

In previous works, we reported results on the damages induced by free radicals to biomaterials [19] and on the peptide–metal interactions [18,20]. To obtain more profound insights into these factors, in the present paper, we describe a SERS investigation on some oligopeptides derived from EAK16 (hereafter Pept1). In particular, their primary structure was modified by substitution of acid and basic amino acids with others having different chain length: Pept2, Glu → Asp substitution (one CH_2_ less); Pept3, Lys → Orn substitution (one CH_2_ less), and Pept4, where both the previous substitutions were made [17]. Before and after gamma-ray irradiation, these peptides were investigated to study their resistance to free radical stress exposure and if eventual structural changes can modify their interaction with metal NPs. The SERS spectra interpretation was also supported by theoretical quantum mechanical computations in the Density Functional Theory framework (DFT). Computations were performed on model systems made up of simplified peptide sequences and silver atoms interacting with different residues along the peptidic chain to obtain adequate band assignments and better identify interactions of the peptides’ distinct groups with the metal surface. Theoretical spectra were analyzed in terms of the Potential Energy Distribution (PED), which was used to estimate the contribution of different vibrational modes to the experimental Raman band intensities.

## 2. Materials and Methods

The examined peptides were Pept1: H_2_N-(Ala-Glu-Ala-Glu-Ala-Lys-Ala-Lys)_2_-CONH_2_, taken as a reference; Pept2, where the Glu-charged residue was substituted by Asp (one CH_2_ group less); Pept3, where the Lys-charged residue was substituted by Orn (one CH_2_ group less); and Pept4, where both charged residues (Glu, Lys) were substituted by Asp and Orn, respectively. Peptides were synthesized as previously reported [17]. Briefly, the peptides were synthesized by using an automated peptide synthesizer via fluorenylmethoxycarbonyl protecting group (FMOC) chemistry, while the cleavage of the peptides and the deprotection of side chains were achieved using trifluoroacetic acid (TFA); the purity of the peptides ranged between 95 and 99%, although small contamination by FMOC used in the synthesis procedure was revealed in the analysis of the solid samples. Conversely, a residual amount of TFA was never found [17,18,19,20].

The silver colloid employed in this work was prepared by following Leopold and Lendl’s method [20,25]. Briefly, 10 mL of a 10^−2^ M AgNO_3_ solution was added dropwise to 90 mL of a 1.6 × 10^−3^ M solution of hydroxylamine hydrochloride containing 3.33 × 10^−3^ M sodium hydroxide. SERS samples were prepared by adding 10 μL of the oligopeptide solution to 490 μL of the silver colloid in order to reach a final concentration of 10^−5^ M; the obtained solution was shaken for 10 s on a vortex mixer (RX3, Velp Scientifica, Usmate Velate, Italy) before SERS measure. No salt was used as an aggregating agent. AgNPs were characterized by metallic plasmons’ resonances in the UV-Vis spectra, showing a maximum at about 405 nm [20]. Although the NPs were partially aggregated, there were no large clusters, and the isolated NP diameter was of ca. 50 nm, as obtained by the TEM and UV-vis analysis (Appendix A).

SERS spectra were collected on a Renishaw Raman InVia model spectrometer equipped with a Leica microscope electrically cooled CCD camera. Samples were excited by using the 532 nm laser line provided by a frequency-doubled Nd:YAG laser and a power of 2.5 mW at the sample. The spectral resolution was set in all cases to 4 cm^−1^. SERS spectra were registered with a total acquisition of 30 s for each SERS spectrum and consisted of 4 scans. The concentration of peptides in the solution was about 10^−5^ M. 

Raman spectra on solid peptides were recorded on a Bruker Multiram FT-Raman spectrometer, equipped with a liquid nitrogen-cooled Ge-diode detector. The spectral resolution was 4 cm^−1^ and 6000, the number of scans for each spectrum (integration time about four hours). The excitation source was an Nd^3+^-YAG laser (1064 nm, about 100 mW laser power at the sample) in the backscattering (180°) configuration.

The intensity ratios were calculated after a curve fitting analysis was performed using GPL software (Fityk 0.9.0 by Marcin Wojdyr) [26] on the original spectra in the 3020–2800 and 1200–1000 cm^−1^ ranges, using the Levenberg–Marquardt algorithm. The curve-fitting procedures’ peak profiles were described as a linear combination of Lorentzian and Gaussian functions [27]. A realistic identification of the peak composition elements and their position was carried out using the second derivative of SERS spectra obtained by a 9-point smoothed moving average function. 

Reactive species generation, which mimics the conditions of endogenous radical stress, was obtained by γ-radiolysis. Gamma irradiation was performed on oligopeptide aqueous solutions using a ^60^Co Gammacell at the dose rate of ~5.0 Gy/min. In radiolysis of diluted aqueous solutions, the energy of the radiation is deposited in water, leading to the formation of three short-lived species: hydrated electrons (*e_aq_^−^*), hydroxyl radicals (•OH), and hydrogen atoms (•H). The experimental conditions can be tuned to control and select the three short-lived species in their reactivity. For example, by saturating with N_2_O (~0.02 M of N_2_O), *e_aq_^−^* are efficiently converted into •OH (k = 9.1 × 10^9^ M^−1^ s^−1^); •OH and •H radicals account for 90% and 10%, respectively, of the reactive species [28]. This condition has been used to model oxidative damage occurring in vivo [29,30] and on many proteic systems [31,32,33].
(1)$$H_{2}O+e^{-}\;\overset{\gamma }{\to }\;e_{aq}^{-}+•OH+•H$$
(2)$$e_{aq}^{-}+N_{2}O+H_{2}O\;\to \;N_{2}+•OH+OH^{-}$$

After gamma irradiation (200 Gy), the peptides were lyophilized, and their Raman and SERS spectra were collected. Lyophilization was performed on a Modulo 4 K Freeze Dryer equipped with an RV8 Rotary Vane Pump (Edwards). The lyophilized product was kept at −80 °C until use.

Regarding quantum mechanical calculations, to reduce excessive computational load, they were performed on model systems mimicking the sequence of Pept1, Pept2, and Pept3. Since the original peptide sequences are made up of two identical moieties plus termini, model systems included only a single moiety, whereas the terminal groups were maintained. Thus, Pept1 was reduced to Pept1-r: H_2_N-Ala-Glu-Ala-Glu-Ala-Lys-Ala-Lys-CONH_2_, Pept2 to Pept2-r: H_2_N-Ala-Asp-Ala-Asp-Ala-Lys-Ala-Lys-CONH_2_, and Pept3 to Pept3-r: H_2_N-Ala-Glu-Ala-Glu-Ala-Orn-Ala-Orn-CONH_2_. In fact, computations on these shortened sequences led to results able to provide reliable comparisons between experimental and theoretical SERS spectra [20], reducing the total compute time to manageable levels. The pH value was about 6.5. Marvin suite by ChemAxon (www.chemaxon.com, last accessed on 29 March 2021) was used to estimate the relative abundance of peptides with different protonation states in the colloidal mixtures. Since the pH of the peptide/Ag–NPs colloidal mixtures was about 6.5, species with +1 charge were largely dominant (>90%).

Ag colloidal surfaces’ interactions were approximated by placing an Ag_2_ cluster at different positions along the peptide chains, including terminal groups. These positions provided the initial guess for quantum mechanical geometry optimization. The use of Ag_2_ (the most straightforward silver cluster possible) is justified by its ability to simulate point-to-point interactions, similar to the geometric arrangements allowed by colloidal particles and its ability to exhibit νAg–Ag vibration in the silver colloidal particles. Consequently, the Ag_2_ cluster can account for directionality and anisotropy in the interactions between oligopeptides and silver particles, reducing the computational burden of a more significant number of heavy atoms [34].

Quantum mechanical calculations were performed in the Density Functional Theory (DFT) framework by the Gaussian09 program [35]. Every complex corresponding to a different position of Ag_2_ along the Pept1-r, Pept2-r, and Pept3-r chains was optimized. Results for Pept1-r, already published, are reported here for comparison [20]. Different geometries were obtained for Pept2-r (see Appendix A, respectively, for the interaction of Ag_2_ with the COO^−^ group, the amide C=O group (two different settings), and the terminal C=O group), and for Pept3-r (see Appendix A for the interaction of Ag_2_ with the same settings as for Pept2-r). The wB97XD functional was employed, using a version of Grimme’s D2 dispersion model [36]. Computations were performed by the correlation-consistent, polarized, minimally augmented basis set, maug-cc-pVDZ, for all atoms except Ag, modeled by the lanl2dz basis set. Based on literature results [37,38], the choice of this particular combination functional/basis set provides a good compromise between speed and accuracy, as required when dealing with large molecular systems. The implicit solvent model approximated the presence of water molecules in SERS experiments. Calculations were performed by the Self-Consistent Reaction Field (SCRF) using the Polarizable Continuum Model (PCM) [39]. Geometry optimizations were carried on in redundant internal coordinates. According to the implementation in Gaussian09, the convergence criterion was met when maximum and root mean square values of forces and next-step displacements were below predefined thresholds. To improve the accuracy of DFT calculations, a tight convergence criterion was used in the Self Consistent Field (SCF) stage, and the number of points used in the numerical integration of the functional was set to the ultrafine level. Thus, all DFT calculations employed the keywords “scf=tight” and “Int(grid=Ultrafine)” [35].

Finally, theoretical Raman spectra were obtained by frequency calculations on the optimized geometries. Frequencies were computed in the limit of the harmonic approximation, using the same basis sets and method as in the geometry optimization steps. All computed frequencies were positive, confirming that optimized geometries correspond to minima on the Potential Energy Surface (PES). Possible anharmonic effects [40] were accounted for by linear fitting of the theoretical frequencies to the experimental ones using the SPESCA program [41]. Scaled frequencies were computed as ν_scaled_ = *a* + *b* ν_calculated_. Values of a and b are reported for each geometry in the Appendix A. Interpretation of the theoretical frequency spectra was performed by the Potential Energy Distribution (PED) analysis of the fundamental vibration modes. The program VEDA carried on the procedure [41,42], allowing for identification of the stretching, bending, and local torsion modes for each computed line. All theoretical and experimental obtained parameters are reported in the Appendix A.

## 3. Results and Discussion

### 3.1. SERS Spectra of the Peptides As-Synthesized

The SERS spectra of Pept2, 3, and 4 are compared with that of Pept1, the original peptide synthesized by Zhang [4], where acid and/or basic residues have been substituted to obtain the other peptides. At the pH of the SERS measurements (about 6.5), as well as in the solid form used to record the FT-Raman spectra, the peptides were mainly with the acid residues (i.e., Glu and Asp) in the deprotonated form, while the basic residues (i.e., Lys and Orn) were protonated. Moreover, the terminal amino group was protonated, giving all peptides a net charge at pH 6.5 of +1 since the terminal carboxylate group is in the amide form. 

Figure 1 shows the SERS spectra of all peptides normalized to the water stretching band at 3200–3400 cm^−1^, and Table 1 reports the band assignment of all peptides, according to literature, and the results of the theoretical calculations on Pept1, Pept2, and Pept3 interacting with an Ag_2_ dimer in different positions and orientations. 

The CH stretching bands in the 3030–2810 cm^−1^ region are always the strongest bands in all SERS spectra. The peak maximum is at about 2934 cm^−1^, with two shoulders at 2970 and 2875 cm^−1^ (Appendix A). Comparing the CH stretching modes (2970–2875 cm^−1^) of the peptides, they appear more intense in Pept4 than Pept1, whereas they are less intense in Pept2 and Pept3 (Figure 1 and Appendix A). This behavior indicates that the intensity of CH stretching modes of SERS spectra does not directly correlate with the acid/basic amino acids chain length but is presumably related to the interaction between CH_2_ groups and their distance and relative orientation with the NPs surface [57]. In fact, the highest relative intensity of the 2930 cm^−1^ band compared with the water band at 3400–3200 cm^−1^ was measured in the SERS spectrum of Pept4 (Figure 1 and Appendix A), i.e., which contains the shortest aliphatic chain in the charged amino acids (Asp and Orn). This last finding is also related to the Full Width at Half Maximum (FWHM) of the CH stretching band: the minimum value was measured in Pept4 at 43 cm^−1^, while the maximum in Pept1 was 49 cm^−1^, thus reflecting the CH_2_ length in charged amino acids (Table 2).

The intensity ratio between the 1060 and 1130 cm^−1^ components of the CH stretching band has been used to study the order degree and packing of aliphatic chains lipids [58] and, therefore, could give some insights into hydrophobic interactions between side chains; in particular, a decrease of the I_1060_/I_1130_ is a marker of a higher disorder degree in CH side chains. This intensity ratio increased in Pept3 (2.0), whereas it decreased in Pept2 and Pept4 (0.9) (Table 2), which have the most disordered packaging of the aliphatic chains.

The bands at about 1450 and 1330 cm^−1^, assigned to the aliphatic chain’s deformation modes, are well visible in the SERS spectra of all peptides (Table 1). The former shows some differences only in Pept4, where the peak is centered at 1442 cm^−1^, and the second derivative shows an additional component at 1454 cm^−1^ (Appendix A). On the contrary, the latter is different in Pept3: a weak band is observed at 1347 cm^−1^, with a shoulder at 1330 cm^−1^ observable in the second derivative spectrum (not shown). Analogously, other bands attributed to the aliphatic chain between 1200 and 850 cm^−1^ showed many similarities between Pept2 and Pept4, while they are different in Pept3. Pept2 and 4 showed peaks at about 1100, 1030, 970, and 940 cm^−1^ (Figure 2 and Figure 3), while the prominent bands in Pept3 are located at about 1125, 1053, and 950 cm^−1^ (Figure 4). These three bands also appear in the SERS spectrum of Pept1 (Figure 5); they could be attributed to the aliphatic chain of glutamic acid, while those described for Pept2 and 4 (i.e., 1100, 1030, 970, and 940 cm^−1^), to aspartate.

Regarding Pept3, the presence of three bands, asterisked in Figure 4, at about 1615, 1000, and 650 cm^−1^, are not significant because they are due to FMOC, used to synthesize the peptides [54].

Regarding the peptide bonds, their SERS bands’ intensities (particularly Amide I) usually decreased compared to carboxylate bands, indicating a different position for the polypeptidic chain regarding the metal surface. Moreover, theoretical results are consistent with this hypothesis: the optimized geometries of Pept2-r interacting with the Ag dimer through the peptidic bond (Table 3) showed a distance of 2.35–2.36 Å between Ag and the carbonyl groups (Appendix A).

Due to the overlap of the high number of components under the Amide I and III bands, significantly when turns and other structures could scatter in the same or nearby regions, it is judicious not to assign these components to a specific secondary structure, but to use them only to check if changes in the overall peptide structure take place.

The most intense SERS band in the Amide I region of Pept2 (Figure 2a) appeared at 1650 cm^−1^, with a shoulder at a higher wavenumber. Using the second derivative spectra, these two components were more clearly visible at 1649 and 1676 cm^−1^ (Figure 2b), but since SERS spectra are recorded in solution, a significant contribution of the 1640 cm^−1^ water band was overlapped and should be considered. Moreover, in the Amide III bands, some components could be identified at 1288, 1255, and 1238 cm^−1^ (see Figure 2b), indicating the presence of different peptides’ secondary structure motifs, compatible with other studies of peptides interacting with NPs [59]. Since this oligopeptide has a prevalent β-sheet secondary structure in the solid phase [17], it is evident that the interaction with NPs notably changed its folding.

Analogously, the presence of different components both in the Amide I and III bands for Pept3 (Amide I 1679 and 1650 cm^−1^; Amide III: 1265 and 1245 cm^−1^) and Pept4 (Amide I: 1672 cm^−1^ with a shoulder at 1647 cm^−1^; Amide III: at 1258 and 1243 cm^−1^) (Figure 3 and Appendix A) suggest that those peptides adopt quite different foldings due to the interactions with the metal. 

The SERS spectrum of Pept2 (Glu →Asp substitution, one CH_2_ less), shown in more detail in Figure 2a, displays a notable enhancement of the bands attributed to carboxylate vibrations than its Raman spectrum, suggesting that carboxylate groups directly interact with the nanoparticles. The most prominent band of carboxylate groups appeared at 1391 cm^−1^ (symmetric stretching mode ν_s_ COO^−^), intensified and red-shifted of about 10 cm^−1^, compared to the Raman spectrum. This band intensification can be attributed to the COO^−^ proximity to the surface and the occurrence of a charge transfer mechanism [60]. This result agrees with the DFT calculation (Table 3), where the interaction between the Ag_2_ dimer and the carboxylate group of Glu residue gives rise to the most stable system.

These spectral changes can also be observed in other SERS bands attributable to COO^−^ vibrations, such as the ν C-COO^−^, δ COO^−^ and δ COO^−^ vibration modes (Table 1). The first gave rise to a band visible at 910 cm^−1^ in the Pept1 and Pept3 spectra (Figure 5; Figure 4, respectively), which shifted toward lower wavenumbers in the Pept2 and Pept4 spectra (900 cm^−1^, Figure 2; Figure 3, respectively) due to the acidic amino acid substitution; the second one was at 654 cm^−1^ in the spectrum of Pept1 (Figure 5), shifted to 663 cm^−1^ in the spectrum of Pept2 (δ COO^−^, Figure 2); the last gave rise to a broad and intense band at about 563 cm^−1^ (δ COO^−^, Figure 5) [58,59]. These last two bands were mixed with amide motions (Table 1), as suggested by the PED analysis results of the different optimized geometries of the Ag_2_–Pept2-r systems (Appendix A). The most stable Pept2–Ag_2_ systems were obtained when the Ag dimer was set close to the carboxylate group of an Asp residue (Table 3). The presence and enhancement of all these COO^−^ bands indicate that these functional groups lie perpendicular or nearly perpendicular to the silver surface. This result agrees with the absence of both δ COO^−^ and ω COO^−^ bands (720 and 620 cm^−1^, respectively), [55,61], since these vibrations are observed when the COO^−^ groups lie parallel to the metal surface.

To better understand the interaction mechanism of Pept2 with the NPs, the second derivative spectrum was obtained (Figure 2b). It displayed a ν_as_ COO^−^ band at 1555 cm^−1^ and two main components of the ν_s_ COO^−^ mode in the 1390–1414 cm^−1^ region, suggesting that not all the COO^−^ groups interact in the same way with the Ag particles. The presence of both types of vibrations allowed us to calculate a value of Δν = (ν_as_ COO^−^ − ν_s_ COO^−^) 164 cm^−1^, corresponding to bridging bidentate coordination [18,62]. Comparing the Pept2 SERS spectrum with its parent Pept1 (Figure 5), a very similar trend can be found regarding the COO^−^–Ag_2_ interaction, except for the notable intensity increase of the 563 cm^−1^ band (COO^−^ wagging) visible in the Pept2 spectrum. Some authors [63] observed that this deformation band’s intensity is sensitive to the incident laser’s polarization; thus, this band’s strength in the Pept2 spectrum could be attributed to the orientation of carboxylate groups toward the Ag nanoparticles.

Theoretical calculations showed that the Ag dimer lies in the same plane as the carboxylate group at a distance of 2.29 Å from the oxygen atom (see molecular geometries in Table 3), suggesting the setting up of bidentate chelation over the closer Ag atom. 

Analogously, Pept3 (Lys → Orn substitution, one CH_2_ less) showed a significant enhancement of the bands attributed to carboxylate vibrations, indicating the direct interaction of this functional group with NPs (Figure 4). Table 3 reports a similar orientation of the Ag_2_ dimer in both peptides when close to carboxylate groups, as indicated by a similar Δν value (174 cm^−1^). In this peptide, the COO^−^ wagging band is feeble and appears at 541 cm^−1^, i.e., red-shifted of about 20 cm^−1^ compared to Pept2. The red shifting can be explained in agreement with the IR spectrum of amino acids, where the deformation band of the carboxylate group was visible at 553 cm^−1^ for Aspartic acid and 536 cm^−1^ for glutamic acid [61]. 

Additionally, the SERS spectrum of Pept4 (Glu → Asp substitution, Lys → Orn substitution, both one CH_2_ less), reported in Figure 3, shows bands attributable to COO^−^ group very similar to those observed for Pept2, except for the absence of the ω COO^−^ at about 560 cm^−1^. In this case, the Δν value is 157 cm^−1^, thus corresponding to bridging bidentate coordination, as observed for Pept2 [18,62].

### 3.2. SERS Spectra of the Peptides after Irradiation

The SERS spectra of the peptides exposed to •OH radical attack were compared with the nonirradiated ones. As a general rule, SERS spectra evidenced fewer differences between irradiated and nonirradiated peptides than what was observed in the solid phase [19] due to the different experimental conditions: i.e., low peptide concentration in aqueous solution and interaction with silver nanoparticles.

#### 3.2.1. Peptide 1

The parent peptide, upon radical exposure, showed many variations on amides and carboxylate bands, together with some changes in the aliphatic chain bands (Figure 5). In more detail, Amides I and III increased after irradiation and were visible at 1677 and 1246 cm^−1^, respectively. Low wavenumber Amide IV (650 cm^−1^) and Amide VI (560 cm^−1^) were affected by irradiation: these complex vibration modes are mainly associated with C=O bending [20]s; therefore, their reduced intensity, as well as that of the Amide I band, may indicate a variation in the orientation and distance between the amide carbonyl group and Ag nanoparticles. These intensity changes may also reflect a variation of the orientation of carboxylic groups whose bending contributes to these low wavenumber bands; in fact, the irradiated SERS spectrum showed an intensity increase in the 1393 cm^−1^ band (symmetric stretching) with a decrease of the 1580–1530 cm^−1^ range (asymmetric stretching, mixed with Amide II and NH_3_^+^ bending). Those changes also reflect on aliphatic chain bands at 1162, 884, and 818 cm^−1^, which reduced their intensity upon irradiation.

#### 3.2.2. Peptide 2

In Pept2 (Figure 2a), the radical attack induced many changes at the secondary structure level: in fact, both Amide I and Amide III bands showed many differences. The former band shifted its maximum from 1650 to 1668 cm^−1^, with a central component of the second derivative spectrum at 1677 cm^−1^ (Figure 2b). The latter showed a principal peak at 1242 cm^−1^ (1253 cm^−1^ before the radical attack), with a shoulder at 1264 cm^−1^ (1285 cm^−1^ before the radical attack), and with an additional component (from the second derivative spectrum, Figure 2b) at 1288 cm^−1^, confirming that the prevailing secondary structure was changed. Another interesting feature is the increase of the relative intensity of the symmetric stretching band of COO^−^ groups at 1391 cm^−1^, while the asymmetric band shifted from 1555 to 1542 cm^−1^. Moreover, the deformation bands at 663 cm^−1^ and mostly 563 cm^−1^ showed a significant decrease as previously observed in Pept1: therefore, similar considerations about the carboxylic group orientation toward Ag nanoparticles could be drawn.

The most intense bands involving aliphatic groups (CH stretching at 2930 cm^−1^, together with the CH bending at 1447 cm^−1^) showed an increased relative intensity without band shifts (Figure 2a and Appendix A). However, Table 2 shows that the I_2930_/I_2870_ ratio, marker of the order degree, and packing of aliphatic chains increased from 3.2 to 3.7. This last effect could be linked to the increased content of β-sheet structure, an ordered structure that may increase the aliphatic chains’ degree order. The decreased I_1060_/I_1130_ ratio further confirms the previous finding of the aliphatic chains’ increased order (Table 2). The mixed modes bands in the 1200–800 cm^−1^ region, most of which involves CH and CC bonds, showed many shifts and intensity changes; however, due to the complexity of these modes (for attributions, see Table 1 and Appendix A), they could not be used to discuss the orientation of aliphatic chains further.

#### 3.2.3. Peptide 3

After the radical attack, the most significant variation in Pept3 was the notable increase in CH stretching bands’ relative intensity at 2930–2870 cm^−1^ (Appendix A) without any variation of the I_2930_/I_2870_ ratio (Table 2). Accordingly, the CH bending band at 1447 cm^−1^ also increased, together with the 1323 cm^−1^ (ω CH_2_); additionally, the bands at 1090 cm^−1^, 945 cm^−1^, 875 cm^−1^, and 837 cm^−1^, all related to skeletal C-C, C-N, and C-H bonds, increased (Figure 4). Carboxylate group bands generally increased too, particularly the ν_s_ COO^−^ at 1394 cm^−1^ and the ν CCOO^−^ at 909 cm^−1^, while the mixed vibration modes involving amides showed a decrease/shift. In fact, the asymmetric stretching mode showed a decrease in the 1575 cm^−1^ component and an increase in the 1551 cm^−1^ one, and the δ COO^−^ band at 658 cm^−1^ almost disappeared, while the ω COO^−^ at 541 shifted to 552 cm^−1^ upon radical attack. All these changes reflect a different orientation of carboxyl groups of the irradiated peptide that can also be affected by variation occurring to the secondary structure adopted by the peptide. Moreover, amides bands showed changes in their position and relative intensity. In the Amide I region, the component at 1677 cm^−1^ (second derivative spectrum, Appendix A) increased, as previously observed in Pept1 and Pept2. Interestingly, the Amide V band at 752 cm^−1^ increased its intensity and shifted to 763 cm^−1^; since this vibrational mode has a significant contribution from the out of plane bending of the amide N-H bond, this change reflects a closer interaction between Ag nanoparticles and this chemical group, as also suggested by the increase of the bands mentioned above at 1090, 945, 875, and 837 cm^−1^ involving the C-N bond, too. This effect may explain the unexpected increase of the I_1060_/I_1130_ ratio (Table 2) that is not correlated to an increased order degree of the aliphatic chains (see the I_2930_/I_2870_ ratio, Table 2).

#### 3.2.4. Peptide 4

This last peptide was poorly affected by the radical attack; thus, it appears as one of the most resistant biomaterials to oxidative stress conditions (Figure 3). Bands attributed to the COO^−^ group showed a slight reduction of their relative intensity (the components of the asymmetric stretching band at 1570 and 1535 cm^−1^, the symmetric stretching band at 1391 cm^−1^, and the C-COO^−^ stretching band at 899 cm^−1^), while the mixed band at 554 cm^−1^ was observed only in the spectrum of the peptide after irradiation (COO^−^ and Amide VI). All the amide bands were poorly affected by irradiation: both peak maxima and the second derivative components remained almost unchanged in the Amide I, III, and V bands. More intense variations affected the aliphatic chains: CH stretching at 2930 cm^−1^ increased in wavenumber and relative intensity than the 2870 cm^−1^ component (Table 2 and Appendix A). The 1125 cm^−1^ band (ν CC and τ HCN) and the CH bending at 1450 cm^−1^ decreased; this last band lost the component at 1442 cm^−1^ after the radical attack (Appendix A). These findings may indicate the Asp residues’ partial decarboxylation, which may have affected the I_1060_/I_1130_ ratio (Table 2).

## 4. Conclusions

The SERS technique allows detection of all the examined peptides at a 10^−5^ M concentration, proving adequate identification in a biological environment. The peptide–Ag colloid interaction is prevalent due to COO- groups, with the peptidic bond tilted and close to the silver surface even after free radical stress exposure of the biomaterials, in agreement with the quantum mechanical data indicating that the most stable optimized geometries for the analyzed peptides are obtained by silver–carboxylate interaction. The variation of charged amino acid in the peptide sequence slightly affected the orientation of carboxylate groups without altering the primary interaction mechanism with the Ag nanoparticles. These interactions, coupled with the aqueous environment, deeply affected the secondary structure adopted by peptides compared to the solid phase.

After the oxidative free radical attack, all the peptides’ spectra indicated a different spatial disposition of the aliphatic chains and an increased aliphatic chain order that also affected the position of the amides bands, sensible to the secondary structure. Two different causes can induce the observed modifications on hydrophobic chains: the self-assembly in a more ordered structure, as occurs for Pept2, or by a partial decarboxylation, as in Pept4.

## Figures and Tables

**Figure 1 biomolecules-11-00959-f001:**
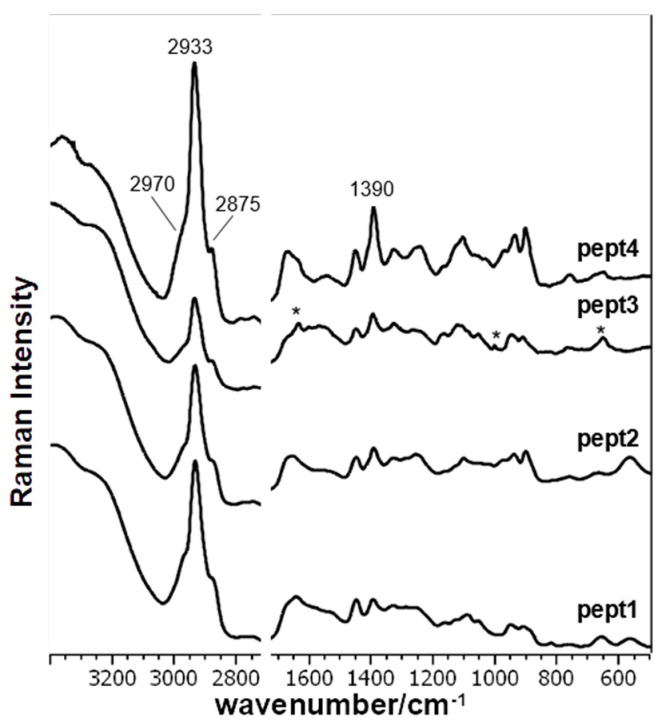
SERS spectra of the examined peptides (Pept2, Pept3, and Pept4) at 10^−5^ M concentration compared with the EAK parent peptide (Pept1). Asterisks (*) were used to indicate the bands attributed to FMOC used in the synthesis procedure.

**Figure 2 biomolecules-11-00959-f002:**
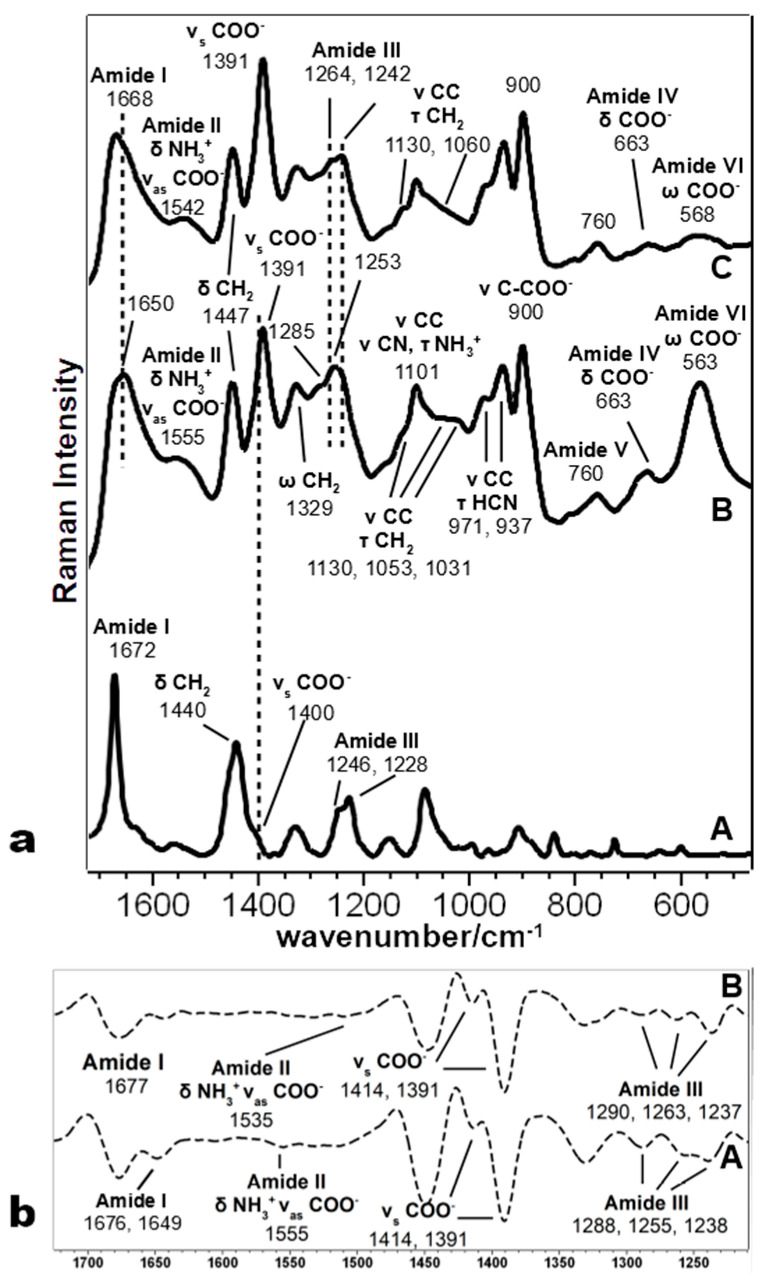
(**a**) Comparison of the Raman spectrum (A) and SERS spectra of Pept2 as-synthesized (B) and after irradiation (C); (**b**) second derivative SERS spectra of Pept2 before (A) and after (B) irradiation.

**Figure 3 biomolecules-11-00959-f003:**
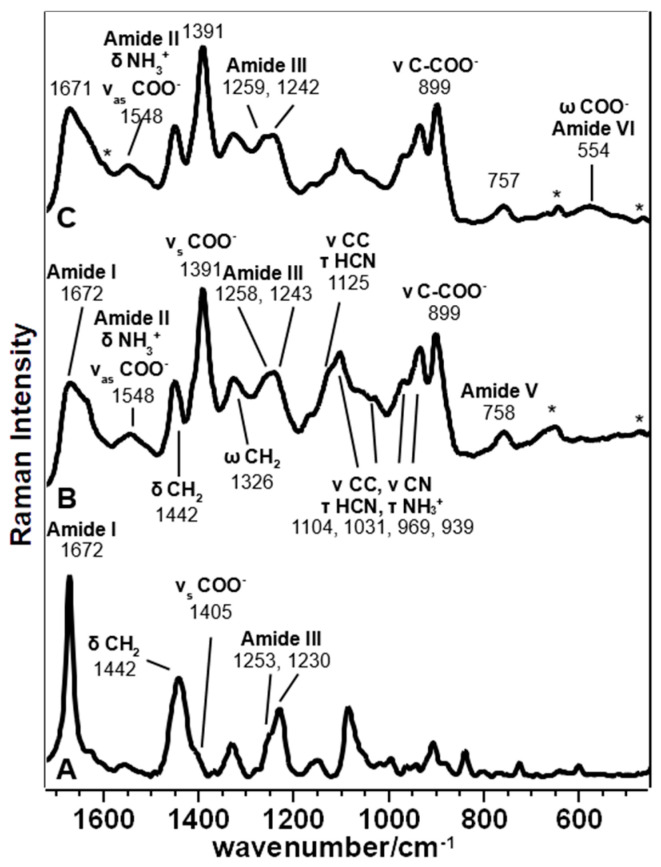
Comparison of the Raman spectrum (A) and SERS spectra of Pept4 as-synthesized (B) and after irradiation (C). Asterisks (*) were used to indicate the bands attributed to FMOC or a contaminant of the solution.

**Figure 4 biomolecules-11-00959-f004:**
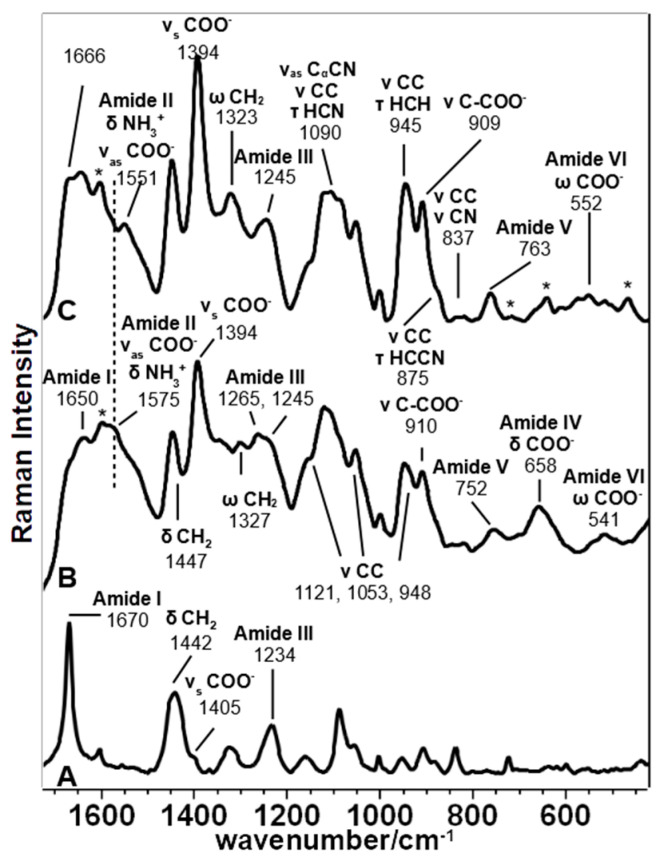
Comparison of the Raman spectrum (A) and SERS spectra of Pept3 as-synthesized (B) and after irradiation (C). Asterisks (*) were used to indicate the bands attributed to FMOC or the solution’s contaminant.

**Figure 5 biomolecules-11-00959-f005:**
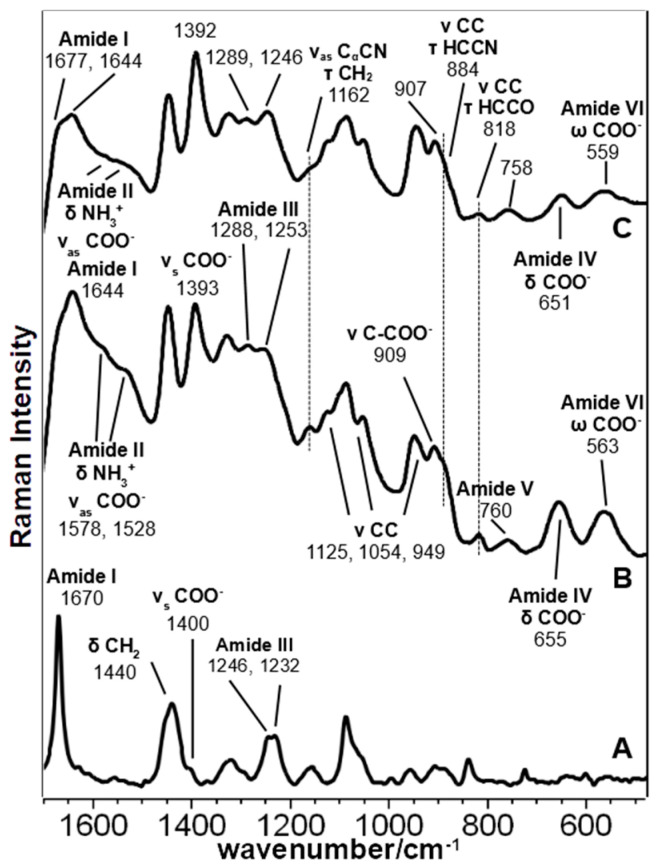
Comparison of the Raman spectrum (A) and SERS spectra of the parent peptide (Pept1) as-synthesized (B) and after irradiation (C).

**Table 1 biomolecules-11-00959-t001:** Attribution of experimental SERS (Surface-Enhanced Raman Scattering) spectra of Pept1, 2, 3, and 4. In the “Assignment” column, the attributions obtained by the PED analysis on the most stable peptide–Ag_2_ geometry is reported (except for Pept4, whose attributions are not supported by PED analysis and are made in agreement with those of the other peptides). These attributions were further confirmed by those present in the literature. (Interpretation of vibrations: ν = stretching, δ = bending, τ = torsion, ω = wagging, ρ = rocking, sh = shoulder, br = broad, vs = very strong, s = strong, m = medium, w = weak, vw = very weak, as = anti-symmetric, s = symmetric).

Assignment	Pept1 H_2_N-Ala-Glu-Ala-Glu-Ala-Lys-Ala-Lys-Ala-Glu-Ala-Glu-Ala-Lys-Ala-Lys-CONH_2_		Pept2 Glu→Asp Substitution		Pept3 Lys→Orn Substitution		Pept4 Glu→Asp and Lys→Orn Substitutions	
	SERS	SERS Gamma 200 Gy	SERS	SERS Gamma 200 Gy	SERS	SERS Gamma 200 Gy	SERS	SERS Gamma 200 Gy
ν CH (aliphatic) [24,43]	2968 sh 2934 vs 2878 sh	2968 sh 2934 vs 2875 sh	2968 sh 2934 vs 2878 sh	2968 sh 2934 vs 2878 sh	2968 sh 2934 vs 2878 sh	2968 sh 2934 vs 2878 sh	2968 sh 2934 vs 2878 sh	2968 sh 2934 vs 2878 sh
Amide I [43,44,45,46,47,48,49]	1679 sh	1677 sh 1670 sh	1674 sh	1668 m, sh 1653 m (1676 + 1649 sh)	1679 sh	1666 m (1677 + 1640 sh)	1672 m (1680 + 1654 sh)	1671 m (1681 + 1656 sh)
δH_2_O [50]	1644 m	1644 m 1638 sh	1650 m		1650 m			
ν_as_ COO^−^ [45] δ NH_3_^+^ [51,52] FMOC [53]					1600 m	1603 m	1602 sh	1603 sh
δ NH_3_^+^ (Lys) [45,47]	1579 sh	1575 sh		1567 sh	1579 sh	1577 sh	1572 sh	1575 sh
ν_as_ COO^−^ [24,44,46] δ NH_3_^+^ (Lys) [54] Amide II [43,48]	1530 sh	1551 sh 1531 sh	1555 br	1542 br	1568 sh 1542 sh 1531 sh	1551 m	1559 sh 1548 br 1535 sh	1559 sh 1548 br
δ CH_2_ [43,44,45,46,50]	1447 m	1447 m	1449 m	1447 m	1447 m	1448 m	1454 sh 1442 m	1450 m
ν_s_ COO^−^ [24,43,44,45,46,47,49,55]	1393 m	1392 m	1414 sh 1391 m	1414 sh 1391 m	1420 sh 1394 m	1394 m	1415 sh 1391 m	1417 sh 1391 m
ω CH_2_(Lys) [24,44,45,54]	1329 m	1325 m	1328 m	1326 m	1327 w	1323 m	1326 m	1326 br
Amide III [44,45,46] ω CH_2_ [54,56]	1288 m	1289 m	1285 sh	1283 sh		1287 sh		
Amide III [43,44,45,46] ω CH_2_ [56]	1253 br	1246 m	1253 br	1264 sh	1265 m		1258 sh	1259 sh
Amide III [44,45,46] ω CH_2_ [56] τ C_α2_H_2_ [44,46]			1238 sh	1242 m	1245 sh	1245 m	1243 m	1242 m 1232 sh
ν_as_ C_α_CN [44,48] τ CH_2_ (Lys) [52,54]	1162 w	1162 sh	1164 sh	1158 sh	1158 sh		1161 w	1155 sh
ν CC [46] τ HCN [24,45,49]	1125 w	1122 w	1131 sh	1123 sh	1121 m	1121 m	1125 sh	1130 sh
τ NH_3_^+^ [24,44,47,49,51,55] ν CC [46] ν CN [49]	1102 sh	1100 sh	1101 m	1100 m	1106 sh	1105 m	1104 m	1101 m
ν_as_ C_α_CN [24,45] ν CC τ HCN [43,49,51]	1086 m	1085 m	1077 sh	1074 sh	1085 sh	1090 sh	1086 sh	
ν CC [49] τ HCH [24,45,51,55] τ CH [51,55]	1054 m	1052 w	1053 vw	1048 sh	1053 sh	1053 m	1055 sh	1056 sh 1031 sh
ρ CH_2_ [44] ν CC [46] ν CN, δ NH_2_ FMOC [53]		997 sh	1018 sh	1027 sh	1001 w	1001 w	1032 sh	1030 sh
ν CC [45,46,49] τ HCH (Lys)	949 m 938 sh	945 m	971 w 937 m	969 sh 935 m	948 m	945 m	969 w 935 m	970 w 934 m
ν COO^−^ [45,46,55]	909 m	907 m	900 m	900 m	910 m	909 m	899 m	899 m
ν CC [44,45] τ HCCN	882 sh	884 sh			885 sh 871 sh	886 sh 875 sh		
ν CC [46,51]		838 sh	840 sh		851 vw	837 vw	835 vw	835 vw
ν CC [46]τ HCCO [49]	816 w	818 vw	812 vw	801 vw	818 vw	818 vw		815 vw
Amide V [46] τ HCCC [51]	760 w, br	759 w, br	760 br	759 br	752 br	763 m	758 w	757 w
δ COO^−^ [24,45,47] ρ CH_2_ [47]			709 sh	701 sh	701 vw	717 vw	707 sh	711 sh
Amide IV [44] δ COO [49,51]	655 br	651 br	681 sh 663 br	663 br	658 br	667 sh 642 vw	669 sh 649 w	669 vw 643 w
ω COO^−^ [45,46,48,51]				609 sh	594 sh	608 w	600 vw	577 vw
Amide VI [47,49,52] ω COO^−^ [44,45,51]	563 br	563 br	563 m, br	563 br	541 vw	552 w		554 w
τ NH_3_^+^ [51]		525 sh			517 vw	517 vw	517 vw	520 sh
τ CN [47]	473 vw	463 vw				465 vw	468 vw	466 vw
δ CN [47]			429 vw	425 vw			414 vw	424 vw

**Table 2 biomolecules-11-00959-t002:** Selected parameters of SERS bands involving CH vibrations before and after exposure to free radical stress (obtained by gamma irradiation): Full Width at Half Maximum (FWHM) of the νCH band, I_2930_/I_2870_ and I_1060_/I_1130_ intensity ratios indicating the order degree of aliphatic side chains of the oligopeptides [58].

Sample	Not Treated			under Oxidative Radical Stress		
	FWHM νCH/cm^−1^	I_2930_/I_2870_	I_1060_/I_1130_	FWHM νCH/cm^−1^	I_2930_/I_2870_	I_1060_/I_1130_
Pept1	49	3.0	1.4	48	3.0	1.4
Pept2	46	3.2	0.9	47	3.7	0.5
Pept3	46	3.2	2.0	46	3.3	2.4
Pept4	43	3.2	0.9	45	3.9	0.6

**Table 3 biomolecules-11-00959-t003:** Comparison of the relative stability for the interactions of the Ag_2_ cluster with different peptide groups. E_et_ is the sum of electronic and thermal energies (kJ mol^−1^). Lower values of ∆E correspond to more favorable interactions.

Peptide-Ag_2_ Conformations	
Pept1-r: Ag_2_/-COO^−^ 1st setting E_et_ = −8,218,083.1	Pept1-r: Ag_2_/-COO^−^ 2nd setting E_et_ = −8,218,064.5
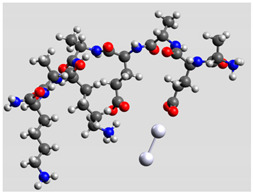 ΔE = 0	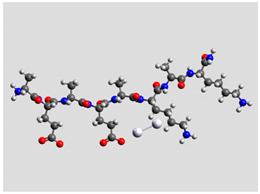 ΔE = 18.64
Pept1-r: Ag_2_/-C=O_chain_ E_et_ = −8,218,058.4	Pept1-r: Ag_2_/-NH E_et_ = −8,218,050.4
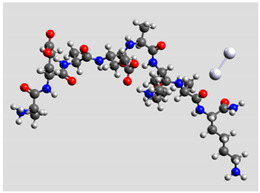 ΔE = 24.67	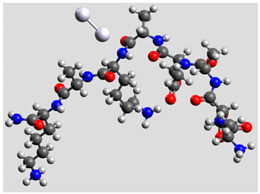 ΔE = 32.70
Pept1-r: Ag_2_/-C=O_terminal_, E_et_ = −8,218,041.1 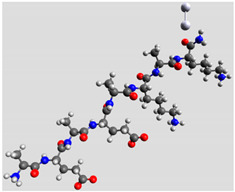 ΔE = 41.99	
Pept2-r: Ag_2_/-COO^−^, (Appendix A) E_et_ = −8,011,715.4	Pept2-r: Ag_2_/-C=O_chain_ 1st setting, (Appendix A) E_et_ = −8,011,703.5
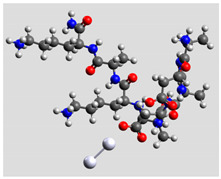 ΔE = 0	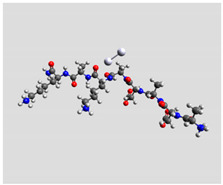 ΔE = 11.96
Pept2-r: Ag_2_/-C=O_chain_ 2nd setting, (Appendix A) E_et_ = −8,011,699.8	Pept2-r: Ag_2_/-C=O_terminal_, (Appendix A) E_et_ = −8,011,696.6
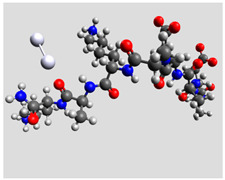 ΔE = 15.64	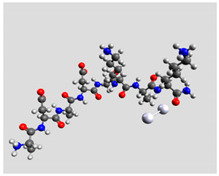 ΔE = 18.81
Pept3-r: Ag_2_/-COO^−^_Glu_, (Appendix A) E_et_ = −8,011,705.0	Pept3-r: Ag_2_/-C=O_chain_ 2nd setting, (Appendix A) E_et_ = −8,011,685.1
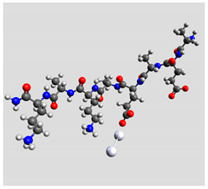 ΔE = 0	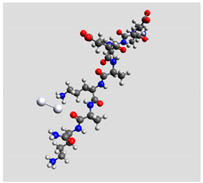 ΔE = 19.98
Pept3-r: Ag_2_/-C=O_chain_ 1st setting, (Appendix A) E_et_ = −8,011,684.7	Pept3-r: Ag_2_/-C=O_terminal_, (Appendix A) E_et_ = −8,011,676.9
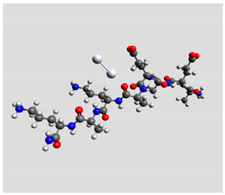 ΔE = 20.30	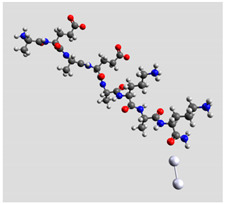 ΔE = 28.16

## Data Availability

The data presented in this study are available in the Appendix A.

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
