# Peer review of "SERS Investigation on Oligopeptides Used as Biomimetic Coatings for Medical Devices"

_biomolecules, 2021, doi:10.3390/biom11070959_

Round 1

Reviewer 1 Report

The manuscript entitled “SERS investigation on oligopeptides used as biomimetric coatings for medical devices” by Foggia et al. reported the use of SERS analysis to study the adsorption of amphiphilic oligopeptides onto silver nanoparticles before and after gamma irradiation.

The manuscript seems well structured and the results agree with the literature data. However, I have some doubts about the SERS study in water solutions if the amphiphilic oligopeptides are supposed to be use as coating materials for medical devices.

I only recommend publication after major revisions.

See observations below:

  1. Why the authors perform this study in aqueous solution? It seems more adequate to perform this kind of studies in solid if these amphiphilic oligopeptides are use as coatings for medical devices. The differences on the amphiphilic oligopeptides after gamma irradiation would be more realist than in aqueous solution. Please comment.
  2. The TEM image and the UV-Vis spectrum of the Ag nanoparticles should be presented at least in the supporting information.
  3. How the authors prepare the SERS samples (amount of Ag colloid; only mix of Ag colloid and Pept or mix for a specific time, etc…). This should be described in the Materials and methods section.
  4. In Page 5, the authors claim that the water stretching band is at 3200-3300 cm-1. In another page (page where is table 2) the authors claim that the same water stretching band is at 3400-3200 cm-1. This should be revised.
  5. How the authors obtain the second derivative SERS spectrum. This should be described in the manuscript.
  6. The Supporting information figures and table should be added in a word document with captions for the readers understand the graphs and tables.

Reviewer 2 Report

In a nutshell, the authors have used surface enhanced Raman spectroscopy to investigate how the presence radical change the conformation of some EAK16 derivatives. They modified the original peptide by replacing E and K with D and O as residues. The latter have shorter side chains. The authors invested time and efforts to produce DFT based vibrational analysis of the peptides. Their spectra indicate that the investigated peptides orient themselves differently with regard to the surface of the silver nanoparticles. This is by no means a surprise. I could not infer any biological useful information from this paper which in my eyes makes it unsuitable for Biomolecules. I suggest that the authors send a revised version to a more specialized journal (Journal of Raman Spectroscopy, Vibrational Spectroscopy). Some of my more specific concerns are listed below.

  1. In the caption of Figure 3 and in the main text the authors assign some peaks in the low wavenumber region to Fmoc modes. This tells me that their peptides have not been sufficiently purified. I wonder whether their samples contained TFA as well which could obfuscate the interpretation of the amide I region.
  2. The authors’ discussion of differences between the amide I profiles in the SERS spectra of the investigated peptides ignores the fact that intrinsic amide I wavenumbers are side chain dependent (JPC B, 109, 8195, 2005). Overall, their amide I analysis is based on the assumption that individual secondary structures give rise to well defined Gaussian amide I bands. However, this is by no means true. Vibrational coupling between amide I modes in oligo- and polypeptides produce a dispersion and thus asymmetric amide I band profiles (BBA 1120,123, 1990; JCP 96, 3379; 1992, JPC B 108, 16965, 2004; JPC B 124, 1703, 2020).
  3. While the peptide concentrations used for the SERS experiments are certainly below the respective critical concentrations for self-assembly I wonder about the concentration used for the FT-Raman experiments. They are nowhere provided. 
  4. The authors justified the use of SERS with the argument that normal Raman spectra of peptides are contaminated by fluorescence. Based on my own experience this is not necessary the case if the peptides have been properly purified. Did the authors ever try to measure Raman spectra with their 532 nm laser line? 
  5. Are the authors aware on the fact that amide I mixes with HOH bending (JPC 99, 3074-3083, 1995)?
  6. The authors’ interpretation of amide III data overlooks the multiplet character of this band ( JACS 126, 8433, 2004 and references cited therein, JPC B 106, 4294, 2002).  

Round 2

Reviewer 1 Report

I believe that the manuscript can be published in the present form

Author Response

We thank the Reviewer for appreciating our efforts in modifying the text following his/her indications.

Reviewer 2 Report

Recommendation: Decline
A reading of the revised version did not change my opinion. The authors responded well to some of my points but even after a second reading I do not see much of a biological relevance. The authors observed some changes in the presence of oxidizing conditions, but the biological significance remains everybody’s guess. I doubt that either amide I or amide III in SERS shall be used for structural interpretation. If certain parts of the investigated peptides are in closer contact with the surface than others local modes get selectively enhanced with some excitonic distribution over adjacent peptide groups. The resulting amide I dispersion could thus be different from what one would observed for the same structure in solution. The authors didn’t really appreciate my comment on amide III. Among the many sub-bands of an amide III, there is only one that is structure sensitive. Are the authors sure that ’s the band dominating the SERS spectrum. One cannot assign different bands in the region between 1200 and 1330 cm-1 to different secondary structures. I urge the authors to read the relevant papers from the Asher group which appeared between 2000 and 2010.

Author Response

We thank the Reviewer for his/her suggestion. We kindly invite him/her to read our response in the attachment.
